# VoiceTuner: Self-Supervised Pre-training and Efficient Fine-tuning For Voice Generation

## ABSTRACT

Voice large language models (LLMs) cast voice synthesis as a language modeling task in a discrete space, and have demonstrated significant progress to date. Despite the recent success, the current development of voice LLMs in low-resource applications is hampered by data scarcity and high computational cost. In this work, we propose VoiceTuner, with a self-supervised pre-training and efficient fine-tuning approach for low-resource voice generation. Specifically, 1) to mitigate data scarcity, we leverage large-scale unlabeled dataset and pre-train VoiceTuner-SSL without pre-defined applications, which can be fine-tuned in downstream tasks; 2) to further reduce the high training cost in complete fine-tuning, we introduce a multiscale transformer adapter to effectively update only around 1% parameters as a plug-and-play module. Experimental results demonstrate that VoiceTuner-SSL presents strong acoustic continuations, and VoiceTuner achieves state-of-the-art results in rich-resource TTS evaluation compared with competitive baseline models. Low-resource (1h, 10h, 30h) downstream applications including zero-shot TTS, instruction TTS, and singing voice synthesis present VoiceTuner's superior audio quality and style similarity with reduced data requirement and computational cost. Audio samples are available at https://VoiceTuner.github.io

## CCS CONCEPTS

• **Applied computing** → **Sound and music computing**; • **Computing methodologies** → **Natural language generation**.

## KEYWORDS

Speech Large Language Models, Efficient Fine-tuning, Text-to-Speech, Singing Voice Synthesis

## 1 INTRODUCTION

Voice synthesis [31, 34, 41] aims to generate human-like voices, which attracts broad interest in the machine learning community. A growing number of applications, such as voice assistant services and long-form reading, have been actively developed and deployed to real-world speech platforms.

Current voice large language models (LLMs) [19, 40, 50] cast voice synthesis as a language modeling task in a discrete representation space. VALL-E [40] proposes a language model approach for text-to-speech (TTS) with audio codec tokens. UniAudio [43]

introduces a multi-scale transformer to enable sub-quadratic self-attention, unlocking better performance at a reduced cost for training and generation. A line of works [1, 3, 19] introduces the hierarchical approach that combines semantic and acoustic audio tokens to decrease supervision in model training.

Despite the success achieved, the current development of voice LLMs in low-resource scenarios (training data with limited labels) is hampered by two major challenges: 1) data scarcity: most existing models rely on web-scale training data, which are lacking in low-resource scenarios; and 2) high computational cost: training voice LLMs from scratch are computationally intensive and time-consuming, and the inefficient attention mechanism in transformer further challenges model in modeling long codec sequence.

In this work, we propose VoiceTuner, with a self-supervised pre-training and efficient fine-tuning approach for low-resource voice generation. To alleviate data scarcity, we pre-train the next-token prediction model (VoiceTuner-SSL) in the large-scale unlabeled dataset, which can be fine-tuned in downstream generation tasks with reduced data and device requirements. To further reduce computational cost and avoid losing the general abilities of VoiceTuner-SSL, we introduce a multiscale adapter with separated fine-tuning strategies for global and local transformers, effectively fine-tuning only around 1% parameters in downstream applications.

VoiceTuner is pre-trained on ~160K hours of unlabeled voice data without supervision, followed by rich or low resource (1h, 10h, and 30h) adaptation in downstream applications including zero-shot TTS, singing voice synthesis, and instruction TTS, respectively generalizing to unseen speaker, modality, and instruction. Experimental results demonstrate that VoiceTuner-SSL keeps acoustic continuations, maintaining speaker identity, emotion, and speaking speed from prompts. VoiceTuner exhibits superior audio quality and style similarity, unlocking the ability to generate voice samples in low-resource scenarios.

The key contributions are as follows:

- We present VoiceTuner, with a self-supervised pre-training and fine-tuning approach to alleviate data scarcity in low-resource applications.
- We introduce a lightweight multiscale adapter with separated fine-tuning strategies for global and local transformers, to efficiently fine-tune only around 1% parameters and reduce the computational cost.
- Experimental results present VoiceTuner-SSL's ability to keep acoustic continuations, and demonstrate VoiceTuner's achieves state-of-the-art results in rich-resource evaluation.
- Low-resource downstream applications including zero-shot TTS, singing voice synthesis, and instruction TTS present VoiceTuner's superior audio quality and style similarity with reduced data requirement and computational cost.

## 2 RELATED WORKS

### 2.1 Generative Voice Models

Text-guided voice synthesis (text-to-speech and singing voice synthesis) typically converts input text into mel-spectrogram (e.g., Tacotron [41], FastSpeech [34]), which is then transformed to waveform using a separately trained vocoder [14, 21], or directly generate waveform from text (e.g., EATS [8] and VITS [20]). In zero-shot scenarios, when the distributions of style prompts deviate from the training data, the quality of the synthesized voice often suffers degradation due to distribution mismatches: Meta-StyleSpeech [29] generally adopts a speech encoding network for multi-speaker synthesis. GenerSpeech [16] leverages multi-level style adaptors for the global and local stylization of the custom utterance. YourTTS [4] is built upon VITS with several novel modifications for zero-shot multi-speaker and multilingual training.

### 2.2 Generative Voice Pre-training and Fine-tuning

Self-supervised learning (SSL) [2, 11] has been shown to achieve remarkable advances in recent years, opening up a wide array of applications that leverage their power by adapting models. AudioLDM 2 [27] leverages AudioMAE [13] and performs self-supervised audio generation learning with a latent diffusion model conditioned on audio tokens. UniAudio [43] trains on different generative tasks to obtain prior knowledge in the inter-relationship between audio and other modalities and support new audio generation tasks after simple fine-tuning. Spear-TTS [50] is pre-trained on a BERT-like [7] denoising pretext task, where the model is provided with a corrupted version of an original semantic token sequence with the goal to produce the corresponding uncorrupted token sequence. Differently, we pre-train LLMs (namely VoiceTuner-SSL) in a next-token prediction task without supervision, which has not been investigated in voice synthesis task.

Efficient fine-tuning aims to reduce data and device requirements in downstream generation tasks. SpeechFlow [26] achieve better performance utilizing low-rank adaptation (LoRA), which adds the linear input projection to each self-attention layer. AudioBox [39] include two-stage full fine-tuning to improve model fidelity and quality where all parameters are optimized together. In this work, we introduce a multiscale transformer adapter for parameter-efficient adaptation, which updates only around 1% of the parameters on top as a lightweight plug-and-play module.

### 2.3 Spoken Language Models

Recent generative models cast voice synthesis as a language modeling task to perform in-context learning: VALL-E [40] uses discrete codes derived from an off-the-shelf neural audio codec model, and regards TTS as a conditional language model. Spear-TTS [50] leverage back-translation and prompt-guided LLMs for high-quality TTS with limited supervision. Jiang et al. [17] train a prosody language model with arbitrary-length speech prompts to produce expressive and controlled prosody. GSLM [23] with "textless NLP" is proposed to model language directly without transcription by training autoregressive generative models of low-bitrate Hubert [11] tokens. AudioLM [3] and MusicLM [1] cast audio synthesis as a language

modeling task and leverage a hierarchy of coarse-to-fine units. However, these existing voice LLMs are trained from scratch using web-scale data, replicating this success is limited in low-resource scenarios.

## 3 METHOD

In this section, we first overview the motivation, and introduce generative self-supervised pre-training with follow-up fine-tuning approach with discrete voice representation. Next, we propose a lightweight, plug-and-play adapter for parameter-efficient fine-tuning. In the following, we introduce the scalable global and local architecture in Section 3.3.

### 3.1 Motivation

Current voice large language models (LLMs) [3, 19, 40, 50] cast voice synthesis as a language modeling task in a discrete representation space. However, these voice LLMs are trained from scratch using web-scale data, replicating this success in low-resource scenarios is hampered by two major challenges: 1) data scarcity: large-scale training data are lacking in low-resource scenarios; and 2) high computational cost: training voice LLMs from scratch are computationally intensive and time-consuming, and the inefficient attention mechanism in transformer further challenges model in modeling long codec sequence.

To alleviate data scarcity, we pre-train the next-token prediction model (VoiceTuner-SSL) in the large-scale unlabeled dataset, which can be fine-tuned in downstream generation tasks with reduced data and device requirements. To address the high computational cost in low-resource scenarios, we propose VoiceTuner to respectively decrease data requirements and learnable parameters in fine-tuning with multiscale transformer adapter.

### 3.2 Speech Representation

Audio codec models such as SoundStream [45] and Encodec [6] have recently shown that encoder-decoder architecture excels at learning acoustic information in a self-supervised manner, where the representation can be used in a variety of generative tasks.

The acoustic codec model typically consists of an audio encoder, a residual vector-quantizer (RVQ), and an audio decoder: 1) The audio encoder $E$ consists of several convolutional blocks with a total downsampling rate of 320 and generates continuous representations at every 20-ms frame in 16kHz. 2) The residual vector-quantizer $Q$ produces discrete representations $a_q$ with a codebook size of $K_2$, using a vector quantization layer [37]. 3) The audio decoder $G$ reconstructs the signal $\hat{y}$, from the compressed latent representation $a_q$. In the end, a speech utterance $y$ is represented as acoustic tokens with $[a_1, a_2, \ldots, a_T]$, $a_i \in \{0, 1, \ldots, K_2 - 1\}, \forall 1 \leq i \leq T$, where $T$ is the number of frames.

### 3.3 Multiscale Architecture

With powerful models, large language models have recently exhibited high-quality samples in natural language processing. To make audio modeling more tractable, recent studies propose to represent audio signals as multiple streams of discrete tokens representing the same signal and flatten these codes [1, 22]. It comes at the high computational cost of modeling extremely long sequences, because of

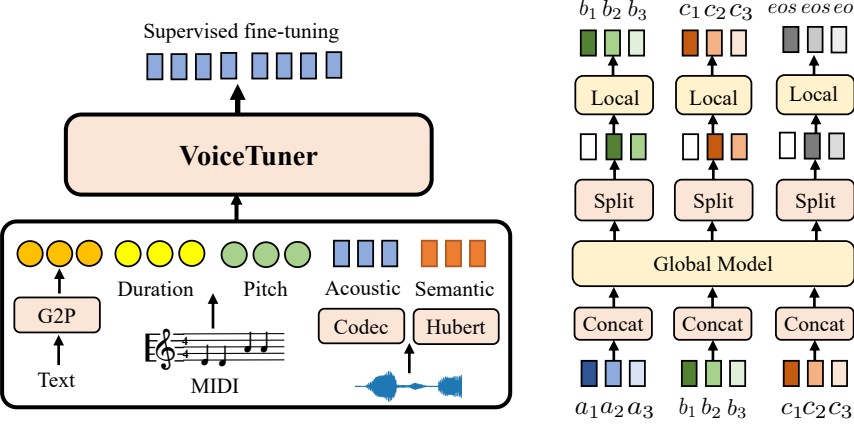

(a) Generative pre-training      (b) Efficient fine-tuning      (c) Multi-scale architecture

**Figure 1: In subfigure (b), prompts can be adjusted for different tasks with a variety of conditions (speaker, emotion, prosody, and style).**

the quadratic cost of self-attention and large feedforward networks per-position.

Our model (denoted as $\theta_{AR}$) predicts long sequences with end-to-end differentiable multiscale transformers or state space models similar to Yang et al. [43], Yu et al. [44]. This enables sub-quadratic calculation, unlocking better performance at reduced cost for both training and generation. As illustrated in Figure 1(c): 1) the token embedding matrix $E_G$ maps integer-valued tokens $a_1, a_2, ..., c_2, c_3$ to $m$ dimensional embeddings, following which 2) we chunk it into patches of size $P$ of length $K = \frac{T}{P}$, 3) a large global model $\theta_{AR}^{global}$ module outputs patch representations $\mathbf{G_o^{1:K}} = \theta_{AR}^{global}(\mathbf{G_i^{0:K-1}})$, and 4) a relatively smaller local model $\theta_{AR}^{local}$ operates on a single patch containing $P$ elements, each of which is the sum of an output from the global model and an embedding of the previous tokens, and autoregressively predict the next patch $\mathbf{L_o^{1:K}} = \theta_{AR}^{local}(\mathbf{L_i^{0:K-1}} + \mathbf{G_o^{1:K}})$.

Our model presents the improvements from scaling attention layers' depth and width without the requirement of scattered model-specific methodologies. As expected, scaling the model size (160M (base), 420M (medium), and 1.1B (large) parameter) results in better scores. We refer the reader to Section 8.1 for our findings.

### 3.4 Self-supervised Pre-training

Most voice LLMs rely on web-scale training data and cast voice synthesis as a language modeling task, while the data shortage hampers its application in low-resource scenarios. To alleviate it, we leverage unlabeled corpus and pre-train LLMs (namely VoiceTuner-SSL) in a next-token prediction task without supervision, where we hypothesize that a generative model without pre-defined application can be applied to different downstream tasks, reducing data requirement in low-resource application.

VoiceTuner-SSL is pre-trained on arbitrary voice, which contains many speakers with various accents, diverse demographics, and

heterogeneous recording conditions. Next, we fine-tune VoiceTuner-SSL to align speech and text modalities utilizing supervised data in downstream voice generation applications, where we find that the self-supervised pre-training stage offers a distinct gain in both rich and low-resource scenarios. We expect our VoiceTuner-SSL to keep the speaker identity, prosody, and recording conditions of the prompt and produce new content. We refer the reader to Section 5 for our findings.

### 3.5 Efficient Fine-tuning

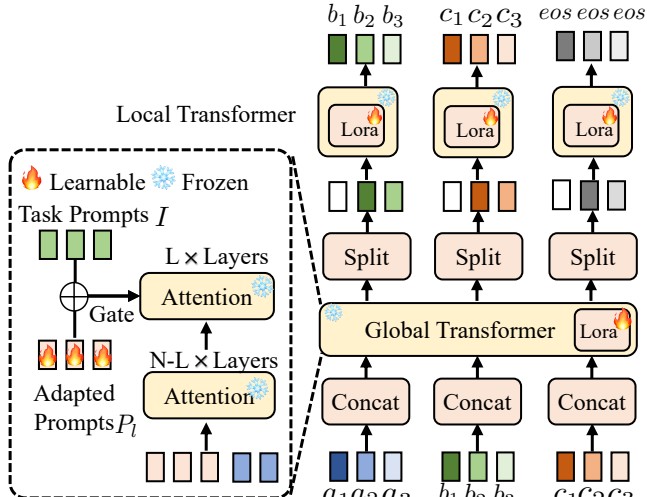

**Figure 2: Efficient fine-tuning with multiscale transformer adapter.**

After pre-training VoiceTuner-SSL on unlabeled speech corpus, we fine-tune the model in downstream tasks with supervised data.

Though fine-tuning voice LLMs is effective compared with training voice LLMs from scratch, a complete fine-tuning of large-scale voice LLMs still 1) is time-consuming, computation-intensive, multi-modality unsupported; and 2) can lose the general ability of foundation model (e.g., acoustic continuations).

In this section, we introduce an efficient plug-and-play module, i.e., a multiscale transformer adapter, where we have separated fine-tuning strategies respectively for global and local transformers to update only around 1% parameters. Specifically,

- In a local transformer, we include low-rank adaptation (LoRA) [12] in the linear input projection of each layer in attention blocks, where only the LoRA parameters are optimized.
- In a global transformer, a set of learnable prompts with gates [49] are added to the input, which learn to adaptively inject new instructions (conditions) into the pre-trained model and avoid disturbing speech tokens at the beginning of training.

Suppose we have condition representation (i.e., task-specific prompts) $I \in \mathbb{R}^{K \times C}$ with length $K$ and feature dimension $C$. For instruction TTS, we use pre-trained Flan-T5-XL [33] and freeze the weights to derive condition representation; For zero-shot TTS and SVS, we use the token embedding matrix to obtain the representation of acoustic and pitch tokens from speaker and MIDI prompt, which are then pad to a fix length $K = 150$.

We initialize learnable adaption prompt $\{P_l\}_{l=1}^{L}$ for $L$ layers, where we have each layer's prompt $P_l \in \mathbb{R}^{K \times C}$ and speech tokens $T_l \in \mathbb{R}^{M \times C}$. Then, the adaption prompt is conducted an element-wise addition with condition representation: $P_l = [P_l + I] \in \mathbb{R}^{K \times C}$.

Suppose the model is processing with the speech tokens $T_l$ and condition $P_l$, The attention score related to learnable prompt is calculated as $S_l^p = \text{Attention}(T_l, P_l, P_l) = \text{Softmax}(T_l P_l^T / \sqrt{C}) P_l$, and we have $S_l^t$ self-attention score for original speech tokens. A learnable gating factor $g_l$ is adapted to adaptively control the importance of $S_l^p$ in the attention with $S_l = S_l^p g_l + S_l^t$, which represents how much information the learnable prompt contributes. Initialized by zero, $g_l$ can first eliminate the influence of under-fitted prompts and then increase its magnitude to provide more instruction semantics.

To conclude, the adaptation enjoys efficient training efficiency with only around 1% learnable parameters. As a lightweight plug-and-play module, this enables us to fine-tune voice LLMs on cheap devices.

## 3.6 Reconstructing High-Fidelity Waveforms

We train a unit-based neural vocoder from scratch for the acoustic unit to waveform generation. Inspired by BigVGAN [24], the synthesizer includes the generator and multi-resolution discriminator (MRD). The generator is built from a set of look-up tables (LUT) that embed the discrete representation and a series of blocks composed of transposed convolution and a residual block with dilated layers. The transposed convolutions upsample the encoded representation to match the input sample rate. Details are included in Appendix B.2.

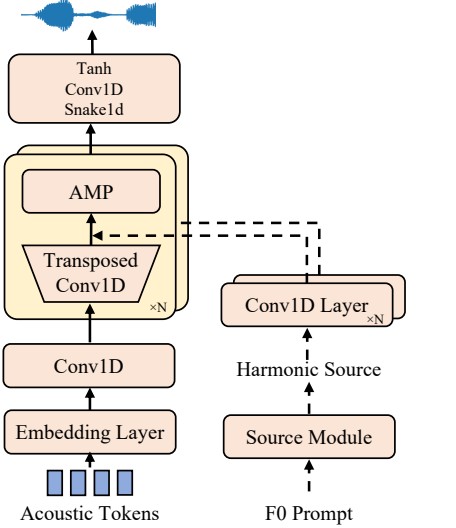

Figure 3: Overview of the unit-based vocoder. The F0 auxiliary input denoted with dotted lines is included only in singing voice synthesis.

## 4 TRAINING AND EVALUATION

### 4.1 Dataset

For self-supervised pre-training, we utilize large-scale datasets with Librilight [18] and WenetSpeech [47], where we have ~**160K** hours of 16 kHz audio that greatly increases the domain coverage.

We fine-tuning VoiceTuner-SSL to align speech and text modalities utilizing TTS data such as LibriTTS [46], VCTK [38] and Prompt-Speech [9], resulting in rich-resource VoiceTuner. For text sequence, we tokenize it into the phoneme sequence with an open-source grapheme-to-phoneme conversion tool [36]. To evaluate Voice-Tuner in low-resource scenarios, we construct paired data (1h, 10h, 30h) with three application tasks: instruction-guided TTS, zero-shot TTS, and singing voice synthesis, respectively generalizing to unseen instruction, speaker, and modality. We have attached detailed information on data configuration in Figure 1.

### 4.2 Evaluation Metrics

**Speech intelligibility.** We report word error rate (WER) to evaluate the intelligibility of speech by transcribing it using a whisper [32] ASR system following [40].

**Style similarity.** SIM assesses the coherence of the generated speech in relation to the speaker's characteristics, and we employ the speaker verification model WavLM-TDNN [5] to evaluate the speaker similarity. F0 Frame Error (FFE) measures the prosody similarity of synthesized and reference audio.

For pitch, speaking speed, and volume attributes accuracy, considering that the values of generated singing may slightly deviate from the boundaries used for categorization, we adopt a soft-margin mechanism for accuracy calculation. Specifically, we take the accuracy of data falling within the correct range as 100, and calculate the accuracy with $100 * \exp(-k\epsilon)$ for data outside the correct range, where $\epsilon$ is the error between the data value and the boundary, and

**Table 1: Dataset usage in self-supervised pre-training and efficient fine-tuning stages. More information is included in Appendix A.**

| Task | Dataset |
|---|---|
| **Self-supervised pre-training** | |
| Next-token prediction | Librilight, WenetSpeech |
| **Rich-resource Evaluation** | |
| TTS (phone/frame level) | LibriTTS, VCTK |
| Zero-shot TTS | Librilight, LibriTTS |
| **Low-resource Evaluation (30/10/1 hr)** | |
| Instruction-guided TTS | PromptSpeech |
| Zero-shot TTS | LibriTTS, VCTK |
| Singing voice synthesis | OpenCPOP, OpenSinger, M4Singer |

$k$ is a hyper-parameter controlling the decay rate of accuracy at the margins, with larger $k$ corresponding to faster decay. We illustrate the accuracy curves in Figure 5 in Appendix C.1.

**Subjective evaluation.** We also conduct a crowd-sourced human evaluation via Amazon Mechanical Turk, which is reported with 95% confidence intervals (CI), and analyze two aspects: style similarity (speaker, emotion, and prosody) and audio quality (clarity, high-frequency), respectively scoring SMOS and MOS. Our subjective evaluation tests are crowd-sourced and conducted by 20 native speakers via Amazon Mechanical Turk on a 1-5 Likert scale.

The MOS (mean opinion score) tests explicitly instruct the raters to "*(focus on examining the audio quality and naturalness, and ignore the differences of style (timbre, emotion, and prosody).)*". For style similarity evaluation, we explicitly instruct the raters to "*(focus on the similarity of the style (timbre, emotion, and prosody) to the reference, and ignore the differences of content, grammar, or audio quality.)*". More information has been attached in Appendix C.

### 4.3 Model Configurations

For acoustic tokens, we train the SoundStream model with 12 quantization levels, each with a codebook of size 1024 and the same downsampling rate of 320. We take three quantization levels as the acoustic tokens, representing each frame as a flat sequence of tokens from the first, second, and third quantization layers. We trained three sets of VoiceTuner, with 160M (base), 459M (medium), and 1.1B (large) parameters. As for the unit-based vocoder, we use the modified V1 version of BigVGAN. A comprehensive table of hyperparameters is available in Appendix B. Except explicitly stated, we use our 459M (medium) model for downstream evaluation.

During training, we pre-train VoiceTuner-SSL for 100K steps using 8 NVIDIA A100 GPUs with a batch size of 6000 tokens for each GPU on the publicly-available *fairseq* framework [30], and fine-tune VoiceTuner for 10K steps using 1 NVIDIA A100 GPU. Adam optimizer is used with $\beta_1 = 0.9, \beta_2 = 0.98, \epsilon = 10^{-9}$. The unit-based vocoder is optimized with a segment size of 8192 and a learning rate of $1 \times 10^{-4}$ until 500K steps using 4 NVIDIA V100 GPUs. For sampling, we employ top-p [10] sampling with p = 0.25.

### 4.4 Baseline

We compare the generated audio samples with other systems, including 1) GT, the ground-truth audio; 2) GT (voc.), where we first convert the ground-truth audio into tokens and then convert them back to audio using BigVGAN; 3) YourTTS [4]: a zero-shot multi-speaker TTS model which is built upon VITS [20]; 4) VALL-E [40] and Spear-TTS [19], recently proposed Speech LLMs for English zero-shot TTS. For easy comparison, the results are compiled and presented in the following sections.

## 5 SELF-SUPERVISED PRE-TRAINING RESULTS

**Table 2: Acoustic continuity of VoiceTuner-SSL.**

| Model | SIM | Emotion | Style | Speed |
|---|---|---|---|---|
| GT | / | 100 | 95.8 | 86.9 |
| GT (voc.) | 0.94 | 93.1 | 92.4 | 87.4 |
| Base | 0.92 | 90.5 | 78.5 | 63.4 |
| Medium | 0.92 | 91.3 | 81.5 | 65.6 |
| Large | **0.93** | **92.7** | **83.1** | **67.1** |

We expect our generative foundation model VoiceTuner-SSL to keep the speaker identity, prosody, and recording conditions of the prompt and produce new content in next-token prediction. Specifically, we generate continuations of 5 seconds for each 3-second prompt, where the prompts are obtained by cropping samples from Librispeech test-clean. In the following, we run the speaker, style, emotion, and speed classifier on the sampled continuations (excluding the prompts) and report the results.

The evaluation results are presented in Table 2, and we have the following observations: 1) VoiceTuner-SSL can preserve the speaker, style, emotion, and speaking speed in the prompt with a high recognition accuracy at a zero-shot setting, even if the model is not fine-tuned in downstream datasets; Informally, VoiceTuner-SSL is optimized in a large amount of self-supervised data, which contains many speakers with various accents and diverse demographics to improve robustness and generalization; and 2) as shown in the demo page, in a noisy environment, VoiceTuner also presents the acoustic consistency and maintain the noise conditions from the prompt.

## 6 RICH-RESOURCE FINE-TUNING RESULTS

### 6.1 Quantitative Findings

Our proposed self-supervised pre-training and follow-up fine-tuning approach are essential for the early-stage training stability and final generation capacity. To demonstrate the rich-resource performance, we fine-tune VoiceTuner-SSL to align speech and text modalities in downstream TTS or frame level TTS (FTTS) tasks, respectively taking phone or duration-expanded phone sequences as inputs.

We plot the loss/accuracy curves in Figure 4 and present results in Table 3, and have the following observations: 1) the model with pre-training converges faster and reaches lower loss bounds than the model trained from scratch; and 2) For the intelligibility of the generated speech, VoiceTuner (with pre-training) has achieved

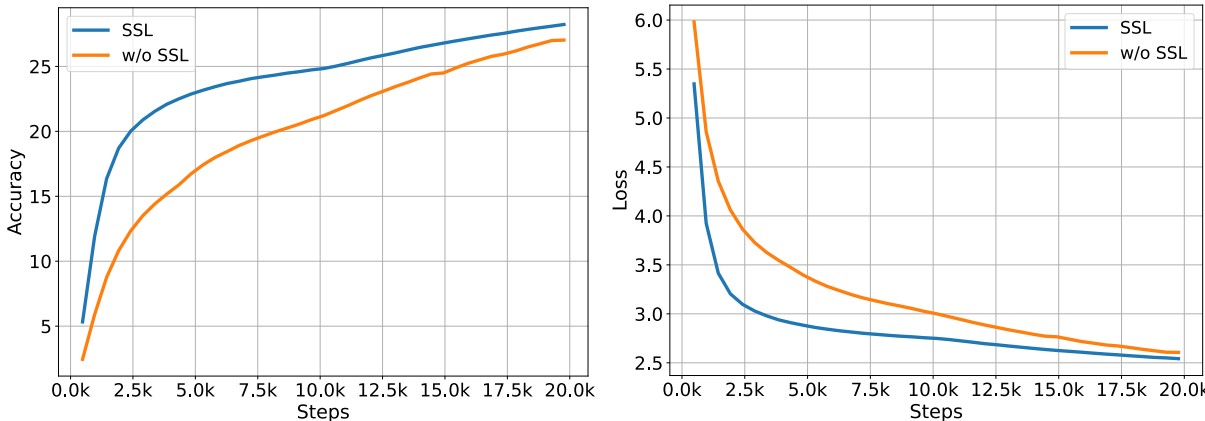

Figure 4: Loss/accuracy curves with or without self-supervised learning (SSL).

Table 3: Low-resource TTS results. FTTS: Frame-level TTS taking expanded phone as input. P: with or without pre-training.

| Task | P | WER | SIM | MOS | SMOS |
|------|---|-----|-----|-----|------|
| GT | / | 3.2 | / | 4.35±0.05 | / |
| GT (voc.) | / | 5.6 | 0.93 | 4.23±0.07 | 4.20±0.05 |
| TTS | ✗ | 9.3 | 0.81 | 3.92±0.07 | 3.84±0.07 |
| | ✓ | **6.7** | **0.83** | 3.98±0.06 | 3.92±0.08 |
| FTTS | ✗ | 6.4 | 0.83 | 3.98±0.07 | 3.93±0.07 |
| | ✓ | **5.9** | **0.84** | 4.04±0.08 | 3.98±0.06 |

a 27%, 7.8% relatively lower WER respectively in TTS and FTTS, indicating that self-supervised pre-training provides gains with accessible speech of better quality. 3) To conclude, VoiceTuner-SSL pre-trained on an arbitrary voice corpus contains speakers with various accents, diverse demographics, and heterogeneous recording conditions, offering distinct gains in rich-resource fine-tuning.

## 6.2 Comparison With Other Models

To compare VoiceTuner with several baselines in the benchmark rich-resource zero-shot TTS tasks, we train our model in the Libri-light dataset and assess the audio quality and style similarity, using a small-scale test set with the examples provided on the demo page. We also score MOS and SMOS for subjective evaluation, rated from 1 to 5, and reported with 95% confidence intervals (CI). The results are compiled and presented in the following table.

1) For audio quality, VoiceTuner has achieved the highest MOS with scores of 4.03 compared with the baseline models; 2) Regarding style similarity, VoiceTuner presents the SIM of 0.85, surpassing baseline models in transferring the style of custom voices. To conclude, self-supervised pre-training on an arbitrary voice corpus offers distinct gains in downstream fine-tuning, and VoiceTuner's direct text-to-acoustic generation avoids the cascaded error

Table 4: Low-resource zero-shot TTS results, we compare VoiceTuner with other models.

| Model | MOS (↑) | SMOS (↑) | WER (↓) | SIM (↑) |
|-------|---------|----------|---------|---------|
| GT | 4.32±0.08 | / | 3.2 | / |
| GT (voc.) | 4.25±0.07 | 4.21±0.06 | 5.6 | 0.93 |
| YourTTS | 3.91±0.07 | 3.81±0.06 | 10.3 | 0.79 |
| VALL-E | 3.92±0.12 | 3.85±0.07 | 8.2 | 0.81 |
| Spear-TTS | 3.97±0.06 | 3.89±0.04 | 7.9 | 0.83 |
| VoiceTuner | 4.03±0.08 | 3.96±0.06 | 6.3 | 0.85 |

in baselines (VALL-E [40]'s cascaded AR and NAR models or Spear-TTS [50]'s cascaded semantic and acoustic tokens). We will include this discussion in the revised version of the paper, and hope this will resolve your concerns.

## 7 LOW-RESOURCE FINE-TUNING RESULTS

We hypothesize that a generative foundation model can be applied to different downstream tasks, reducing data requirements and computational cost, especially in low-resource scenarios. Though it would be more helpful to investigate the real low-resource languages, it would be challenging to train a low-resource language model due to 1) the lack of stable lexicon or grapheme-to-phoneme (G2P) tools, and 2) the expensive and substantial amount of manpower needed for collecting and labeling data, which are beyond our scope. As such, we choose English/Chinese as the target languages, and construct the 1/10/30 hours data to simulate the low-resource languages following previous low-resource speech systems [25, 35].

To present the capability of VoiceTuner in low-resource scenarios, we construct (1h, 10h, 30h) hours of data for three application tasks: instruction-guided TTS (ITTS), zero-shot TTS (ZS-TTS), singing voice synthesis (SVS), respectively generalizing to unseen instruction, speaker, and modality. For training efficiency, we investigate full training from scratch (Full-s), full fine-tuning from VoiceTuner-SSL (Full-p), and efficient fine-tuning with a multiscale transformer adapter (Adapter).

**Table 5: Low-resource instruction TTS results. Full-s: Full parameter training from scratch; Full-p: Full parameter fine-tuning from pre-trained VoiceTuner-SSL. Note that we use / to represent that the model (Full-s) cannot converges in low-resource scenarios.**

| | Gender (↑) | Speed (↑) | Pitch (↑) | Volume (↑) | WER (↓) | MOS(↑) | SMOS(↑) |
|---|---|---|---|---|---|---|---|
| GT | 96.6 | 86.9 | 86.9 | 78.9 | 5.1 | 4.31±0.05 | / |
| GT (voc.) | 95.8 | 87.4 | 87.0 | 76.0 | 7.1 | 4.20±0.07 | 4.20±0.05 |
| **Fine-tune with 30-hour data** | | | | | | | |
| Full-s | 94.1 | **88.3** | **88.2** | 63.9 | 16.9 | 3.94±0.06 | 3.89±0.08 |
| Full-p | **94.7** | 86.1 | 87.3 | **68.3** | 7.1 | 4.01±0.08 | 3.97±0.07 |
| Adapter | 89.1 | 85.1 | 86.7 | 58.8 | **6.9** | 3.96±0.06 | 3.92±0.07 |
| **Fine-tune with 10-hour data** | | | | | | | |
| Full-s | 90.1 | 76.5 | 85.7 | 61.1 | 68.7 | 3.90±0.08 | 3.86±0.08 |
| Full-p | **91.6** | **85.7** | 85.6 | **62.2** | 7.6 | 3.97±0.08 | 3.92±0.08 |
| Adapter | 86.1 | 83.5 | **86.3** | 62.1 | **7.5** | 3.91±0.06 | 3.85±0.07 |
| **Fine-tune with 1-hour data** | | | | | | | |
| Full-s | | | / | | | | |
| Full-p | 49.1 | **84.5** | 77.3 | 57.3 | 14.9 | 3.91±0.08 | 3.84±0.08 |
| Adapter | **80.0** | 82.9 | **85.1** | **61.3** | **9.6** | 3.87±0.06 | 3.82±0.07 |

## 7.1 Zero-shot Text-to-Speech

**Table 6: Low-resource zero-shot TTS results.**

| Model | WER (↓) | SIM (↑) | MOS (↑) | SMOS (↑) |
|---|---|---|---|---|
| GT | 3.2 | / | 4.32±0.08 | / |
| GT (voc.) | 5.6 | 0.93 | 4.25±0.07 | 4.21±0.06 |
| **Fine-tune with 30-hour data** | | | | |
| Full-s | 10.3 | 0.63 | 3.88±0.06 | 3.75±0.06 |
| Full-p | **7.2** | **0.71** | 4.03±0.07 | 3.94±0.08 |
| Adapter | 7.9 | 0.63 | 3.98±0.05 | 3.90±0.07 |
| **Fine-tune with 10-hour data** | | | | |
| Full-s | | / | | |
| Full-p | **8.1** | **0.64** | 3.96±0.06 | 3.91±0.08 |
| Adapter | 8.2 | 0.62 | 3.94±0.07 | 3.89±0.07 |
| **Fine-tune with 1-hour data** | | | | |
| Full-s | | / | | |
| Full-p | 8.9 | 0.58 | 3.90±0.07 | 3.85±0.07 |
| Adapter | **8.5** | **0.60** | 3.89±0.06 | 3.85±0.07 |

In this section, we fine-tune VoiceTuner in a zero-shot TTS task, where we generate the speech conditioned on the acoustic tokens of the 3-second enrolled recording and the phoneme prompt, which constrain the speaker and content information respectively. The results in the zero-shot TTS task are presented in Table 6, and we have the following observations:

1) As training data is reduced in the low-resource scenario, a distinct degradation in speech quality and similarity could be witnessed. For example, VoiceTuner (Adapter) presents a distinct drop in TTS WER of 6.9 → 7.5 → 9.6 when reducing training data from 30 to 1 hours. 2) As the amount of trainable parameters is decreased to 1% in multiscale adapter, only a slight quality drop is observed. As such, effective fine-tuning with the proposed multiscale adapter enables us to fine-tune voice LLMs on limited data and cheap devices. 3) It is worth mentioning that in extremely low-resource scenarios, VoiceTuner (Full-s) cannot converge when training from scratch. As expected, a generative model (namely VoiceTuner-SSL) without a pre-defined application can be applied to different downstream tasks, reducing data requirements in low-resource applications.

## 7.2 Instruction Text-to-Speech

In this section, we fine-tune VoiceTuner in the instruction-guided TTS task, where we take a prompt with both style and content descriptions as input to synthesize the corresponding speech. In this way, users are able to create speech from a prompt, resulting in style control without the requirements for acoustic knowledge or reference speech. The results are presented in Table 5, and we have the following observations:

1) With 30-hour training data, VoiceTuner (full-p) presents high perceptual quality with outperformed subjective and objective evaluation results, which shows that the model can synthesize speech in a consistent style with the intention of style prompts. 2) In extremely low resource scenarios (i.e., 1-hour data), Voice-Tuner (Adapter) presents an outperformed quality compared to full-parameter fine-tuning models (Full-p), and VoiceTuner (Full-s) even cannot converge when training from scratch. To conclude, a complete fine-tuning of large-scale voice LLMs can lose the general ability of the foundation model (e.g., gender continuations), especially in low-resource scenarios. 3) Speech volume is relatively less distinguishable than gender and speed attributes, and thus the

speaking volume classifier presents an accuracy of 78.9% in ground truth data. which is lower than gender (96.6%) and speed (86.9%).

## 7.3 Singing Voice Synthesis

**Table 7: Low-resource instruction SVS results.**

|  | FFE (↓) | SIM (↑) | MOS (↑) | SMOS (↑) |
|---|---|---|---|---|
| GT | / | / | 4.12±0.06 | / |
| GT (voc.) | 0.01 | 0.95 | 4.08±0.04 | 4.02±0.06 |
| **Fine-tune with 30-hour data** | | | | |
| Full-s | | | / | |
| Full-p | **0.31** | **0.93** | 3.97±0.06 | 3.92±0.05 |
| Adapter | 0.43 | 0.90 | 3.93±0.05 | 3.88±0.07 |
| **Fine-tune with 10-hour data** | | | | |
| Full-s | | | / | |
| Full-p | 0.47 | **0.91** | 3.93±0.07 | 3.89±0.06 |
| Adapter | **0.44** | 0.88 | 3.91±0.06 | 3.85±0.08 |
| **Fine-tune with 1-hour data** | | | | |
| Full-s | | | / | |
| Full-p | **0.58** | **0.83** | 3.85±0.07 | 3.69±0.06 |
| Adapter | 0.61 | 0.78 | 3.84±0.08 | 3.71±0.07 |

In this section, we fine-tune VoiceTuner in the singing voice synthesis task, where we generate the singing voice conditioned on the acoustic tokens of the 3-second enrolled recording, F0 prompt (from MIDI), and the phoneme prompt, which constrain the speaker, pitch, and content information respectively. Detailed information on the MIDI-to-F0 converter has been included in Appendix B.1. The results are presented in Table 7, and we have the following observations:

1) In all data usage settings, VoiceTuner (Full-s) cannot converge when training from scratch. Besides, a distinct quality drop can be witnessed when decreasing data usage, which is more significant than those in zero-shot TTS or instruction TTS tasks. To sum up, singing voice synthesis resembles the prosodic style of the F0 prompt and requests a precise pitch reconstruction, and thus it can be more sensitive to data scarcity in low-resource scenarios; and 2) Regarding computational cost, though full parameter fine-tuning systems demonstrate better results in most cases, the multiscale transformer adapter has still achieved the comparable results (e.g., FFE and SIM of 0.61, 0.78 in 1-hour SVS). It indicates that the adapter enjoys high-fidelity generation with only around 1% learnable parameters, which enables us to fine-tune voice LLMs on cheap devices;

## 8 ANALYSIS AND ABLATION STUDIES

To verify the capabilities of VoiceTuner, we conduct ablation studies on model scalability and few-shot adaptation, and discuss key findings as follows.

## 8.1 Scalability to improve performance

As illustrated in Table 8, we report results for different model sizes, namely 160M (base), 459M (medium), and 1.1B (large) parameter

**Table 8: We compare VoiceTuner among different sizes (Base, Medium, and Large).**

| Size | Params | Mem | TFLOPs | WER | SIM |
|---|---|---|---|---|---|
| B | 160M | 4332M | 76.3 | 7.8 | 0.81 |
| M | 459M | 5259M | 181.4 | 6.7 | 0.83 |
| L | 1B | 5638M | 408.1 | 5.9 | 0.84 |

models. As expected, scaling the size of VoiceTuner results in better scores. However, this comes at the expense of longer training and inference time. Increasing the model size from 459M to 1.1B leads to additional gains of a further 40% reduction in WER for TTS tasks with a similar style.

## 8.2 Efficient fine-tuning with multiscale transformer adapter

**Table 9: Ablation studies. We obtain VoiceTuner in low-resource (10-hour) instruction TTS task and report attributes accuracy and WER.**

| Tuning | Params | Gender | Speed | Pitch | WER |
|---|---|---|---|---|---|
| GT | / | 96.6 | 86.9 | 86.9 | 5.1 |
| Lora | 8.97M | **86.6** | 83.2 | 85.8 | 7.6 |
| Adapter | 12.0M | 86.1 | **83.5** | **86.3** | **7.5** |

To enable few-shot learning without losing the general abilities, we fine-tune VoiceTuner in 10-hour instruction TTS data, and compare the results among different adaptation methods. Illustrated in Table 9, as a lightweight plug-and-play module, the proposed multiscale transformer adapter enjoys superior training efficiency with only around 1% parameters in contrast to full fine-tuning, demonstrates the 9.2% WER drop and outperformed attributes accuracy (gender, speed, and pitch) compared to Lora [12]. This enables us to fine-tune voice LLMs on cheap devices.

## 9 CONCLUSION

In this work, we propose VoiceTuner with a pre-training and efficient fine-tuning approach for low-resource voice generation. To mitigate the data scarcity and high computational cost for training voice LLMs, we 1) leveraged large-scale unlabeled dataset and pre-trained VoiceTuner-SSL in a next-token prediction task, which could be fine-tuned in downstream tasks with reduced data; 2) introduced an efficient multiscale transformer adapter to fine-tune only around 1% parameters in downstream applications, further eliminating the computational cost. Experimental results demonstrated that VoiceTuner-SSL presented strong speech continuations. VoiceTuner achieves state-of-the-art results in rich-resource TTS evaluation compared with competitive baseline models. VoiceTuner exhibited superior quality and style similarity with reduced data requirement and computational cost in three low-resource (1h, 10h, 30h) voice generation tasks, including zero-shot TTS, instruction TTS, and singing voice synthesis. We envisage that our work serves as a basis for future low-resource voice synthesis studies.

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
