# OpenReview forum: "VoiceTuner: Self-Supervised Pre-training and Efficient Fine-tuning For Voice Generation"
_acmmm.org/ACMMM/2024/Conference — MM2024 Poster_

### Official Review · Reviewer_WrQu · 2024-05-18

**Rating:** 4
**Confidence:** 3

**Summary:**

This paper attempts the task of voice generation. Specifically, the authors propose a two-stage approach, named as VoiceTuner, encompassing a stage of self-supervised pre-training and a stage of fine-tuning. The first stage takes advantage of the abundance of unlabelled speech data to improve the model performance on low-resource scenarios. In the second stage, the authors propose using both LoRA and learnable prompts for efficient fine-tuning. The results demonstrate VoiceTuner could achieve the state-of-the-art performance for low-resource zero-shot TTS. Through the evaluation on zero-shot text-to-speech, instruction text-to-speech, singing voice synthesis, the authors show that proposed multi scale-transformer adapter can achieve competitive performance comparing to full-fine tuning with very few learnable parameters.

**Strengths:**

1. The paper is well-written and easy to follow.
2. The results are strong with better performance than several baselines. Comprehensive ablation studies have been conducted to demonstrate the effectiveness of proposed efficient fine-tuning strategy.
3. The paper shows that the pipeline of pre-training and fine-tuning auto-regressive models in LLMs can also work in the speech domain.

**Limitations:**

1. The authors only include the comparisons with previous models using a "small-scale test set with the examples provided on the demo page". I think the evaluation is limited and it is better to include more comparisons to justify the effectiveness of VoiceTuner.
2. The proposed pipeline is of limited novelty, which I think is a minor weakness considering that showing a LLM pipeline can also work in speech domain is also valuable. Moreover, the proposed efficient fine-tuning seems to be a combination of learnable prompts and LoRA, which are not new to me.
3. A minor question is that the arrangement of Table 9 is a little confusing. Please correct me if I am wrong, multi-scale transformer adapter is equal to adapted prompts + Lora. So "Lora" in the third row means only using Lora without adapted prompts, right?

**Suitability:**

3

---

### Official Review · Reviewer_HeUq · 2024-05-24

**Rating:** 6
**Confidence:** 2

**Summary:**

The paper introduces a novel approach utilizing self-supervised pretraining alongside parameter-efficient fine-tuning to address data scarcity in voice synthesis. It features a lightweight multi-scale adapter, enabling fine-tuning with just 1% of the model's parameters. The experiments demonstrate state-of-the-art (SOTA) performance across both resource-rich and resource-poor settings, assessed through objective and subjective methods. The model is adept at handling a variety of low-resource downstream tasks such as zero-shot TTS, instructional TTS, and singing voice synthesis.

**Strengths:**

- The paper conducts extensive experiments. Both the effect of self-supervised pretraining and finetuning was demonstrated. The proposed model was compared with SOTA TTS models, including YourTTS, VALL-E, and Spear-TTS. Ablation studies were conducted to confirm the advantage of design choices. The advantage of computation cost is also confirmed by comparing with the full-finetune counterpart.
- The proposed model achieves strong performance in both well-resourced and limited-resource environments, shown by both objective and subjective metrics.
- The analysis and discussion of the result is comprehensive.
- The fine-tuning phase is innovatively designed to incorporate various types of control signals, including MIDI for pitch and duration, as well as audio for acoustic and semantic information.
- The paper is well-organized and clearly written, making it accessible and easy to comprehend.

**Limitations:**

The reviewer has not identified any major issues within the paper.

**Suitability:**

3

---

### Official Review · Reviewer_dD8H · 2024-06-03

**Rating:** 4
**Confidence:** 3

**Summary:**

This paper propoese a self-supervised pre-training and fine-tuning method for low-resource voice generation: 1) large scale unlabeled data are used for model pre-training; 2), a multi-scale transformer adapter is introduced to update a small part of parameters as a plug-and-play modeule. Experimental results show significant improvement achieved.

**Strengths:**

1. speech pre-training and fine-tuning mehtod proposed to alleviate data scarcity in low resource speech generation.
2. a ligntweight multiscale adapater is used for global and local transformers.
3. experiments are conducted for zero-shot TTS, instruction text-to-speech and singing voice synthesis.

**Limitations:**

1. pre-training and fine-tuning methodology is not novel in this area.
2. local and global network structure is not novel for speech generation, as already been used in uniaudio.
4. Is table 4 the result of rich-resource fine-tuining resutls? I think the title of table 4 is wrong.
5. Are the results of other models, such as YourTTS, VALL-E and Spear-TTS from their papers or based on your re-implementation ?
6. What is the result compared with diffusion based method, such as voicbox?
7. The resutls of other models are not compared in the low-resouce setting, why is that?

**Suitability:**

3

---

### Official Review · Reviewer_QyQq · 2024-06-04

**Rating:** 3
**Confidence:** 4

**Summary:**

The paper proposes VoiceTuner, a method for voice generation. It addresses data scarcity and high computational cost using self-supervised pre-training and efficient fine-tuning. VoiceTuner achieves state-of-the-art results in TTS and exhibits superior audio quality in low-resource applications.

**Strengths:**

This paper introduces a pretrain+finetune pipeline for voice generation tasks, which has shown success in the field of large language models (LLMs). The techniques presented in the paper, such as self-supervised pre-training and efficient fine-tuning, offer valuable insights for advancing voice synthesis in the audio community.

**Limitations:**

Missing comparison and discussion to related work such as SpeechFlow, Spear-TTS, and Audiolm.

In Audiolm, the authors propose exploring both semantic tokens and acoustic tokens for pretraining. They find that using only acoustic tokens leads to inconsistencies in linguistic content. However, the authors of this paper still employ acoustic tokens for pretraining. I am curious to know if this approach of using only acoustic tokens with the multi-scale transformer is a better method for audio pretraining compared to the combination of semantic and acoustic tokens. Specifically, I would like to know if your pretraining method can outperform Audiolm in terms of audio quality and long-term consistency for unconditional audio generation.

In Spear-TTS, they utilize a semantic-to-acoustic pretraining approach similar to Audiolm. They also achieve impressive results using only 15 minutes of labeled data and claim to address low-resource TTS. Therefore, I believe it would be valuable for the authors to compare their approach with Spear-TTS under low-resource scenarios.

Additionally, SpeechFlow also adopts a pretrain+finetune pipeline and also utilizes techniques such as LORA and full parameter fine-tuning. It would be beneficial for the authors to discuss the advantages of their strategy compared to SpeechFlow.

Including a comparison and discussion with these related works would provide a comprehensive analysis of the proposed method's effectiveness and highlight its unique contributions in the field of audio pretraining and fine-tuning.

**Suitability:**

3

---

### Meta-Review · Area_Chair_FR88 · 2024-07-01

**Recommendation:** Accept (Poster)
**Confidence:** 4

**Metareview:**

The paper presents VoiceTuner, a method leveraging self-supervised pre-training and efficient fine-tuning for voice generation. Reviewers appreciate that "the paper shows the pipeline of pre-training and fine-tuning auto-regressive models in LLMs can also work in the speech domain" and that "The techniques presented in the paper, such as self-supervised pre-training and efficient fine-tuning, offer valuable insights for advancing voice synthesis in the audio community." Despite concerns about novelty in methodology and comparison breadth, the overall contribution to advancing voice synthesis is acknowledged by most reviewers.